# The Risk Factors of Dewclaw Dermatitis in Beef Cattle in the Amazon Biome

**DOI:** 10.3390/ani14091329

**Published:** 2024-04-29

**Authors:** José Diomedes Barbosa, Janayna Barroso dos Santos, Hanna Gabriela da Silva Oliveira, Tatiane Teles Albernaz Ferreira, José Alcides Sarmento da Silveira, Camila Cordeiro Barbosa, Marilene Farias Brito, Natalia da Silva e Silva Silveira, Carlos Magno Chaves Oliveira, Henrique dos Anjos Bomjardim, Felipe Masiero Salvarani

**Affiliations:** 1Instituto de Medicina Veterinária, Universidade Federal do Pará, Castanhal 68740-970, PA, Brazil; diomedes@ufpa.br (J.D.B.); barrosojanavet@gmail.com (J.B.d.S.); hnnagabriela@gmail.com (H.G.d.S.O.); tatyalbernaz@ufpa.br (T.T.A.F.); jalcides@ufpa.br (J.A.S.d.S.); camilabarbosamedvet@gmail.com (C.C.B.); nataliasilvasilveira1@gmail.com (N.d.S.e.S.S.); cmagno@ufpa.br (C.M.C.O.); 2Departamento de Epidemiologia e Saúde Pública (DESP), Instituto de Veterinária (IV), Universidade Federal Rural do Rio de Janeiro (UFRRJ), Seropédica 23890-000, RJ, Brazil; mfariasbrito@uol.com.br; 3Faculdade de Medicina Veterinária, Instituto de Estudos do Trópico Úmido, Universidade Federal do Sul e Sudeste do Pará (Unifesspa), Xinguara 68557-335, PA, Brazil; henriquebomjardim@unifesspa.edu.br

**Keywords:** foot disorders, production systems, cattle breeding, Amazon

## Abstract

**Simple Summary:**

Beef cattle farming is an extremely important economic activity in Brazil, mainly in the northern region. Bovine hoof disorders impair the beef production chain; however, there are few studies on the subject, especially on the incidence of Bovine Dewclaw Dermatitis (BDCD). Thus, this is a pioneering investigative study to highlight the main factors predisposing animals to BDCD in beef cattle herds in the Amazon Biome. This study shows that risk factors related to climate, pasture conditions, and the physical structure of bovine management facilities are directly linked to BDCD onset in herds, affecting the beef production sector.

**Abstract:**

Bovine Dewclaw Dermatitis (BDCD) is a hoof disease characterized by inflammation of the second and fifth accessory digits and the skin in this region. This pathology is poorly described in the literature; however, it has recently been observed in beef cattle in the Amazon Biome, Brazil. The objective of this study was to perform a clinical diagnosis and identify the risk factors associated with BDCD onset in cattle in the studied biome. Samples were collected from eight farms with extensive breeding systems located in Xinguara, Rondon do Pará, Curionópolis, and Ipixuna do Pará in the state of Pará, Brazil. A total of 706 Nellore and Nellore crossbred with taurine bovine of both sexes were evaluated, with males aged between 2 and 4 years and a mean weight of 650 kg, and females aged between 2 and 11 years and a mean weight of 400 kg. Distal extremities were inspected during cattle management, and in cases of dewclaw lesions, a specific examination was carried out after proper restraint. Cattle were diagnosed with BDCD on all farms analyzed. Of the 706 cattle inspected, 49 (6.94%) showed BDCD, of which 19 (38.77%) were Nellore and 30 (61.22%) were crossbred. This was the first study to determine BDCD’s occurrence in extensive farming systems in the Amazon region, also showing that pastures with large amounts of stumps and stones, the physical structure of pens, and trauma and injury incidence during animal management are the most important predisposing factors for the onset of BDCD.

## 1. Introduction

Brazil is in a prominent position in global agribusiness, mainly in meat production, and has one of the largest cattle herds, with 224,602,112 head. The country stands out as a major global meat exporter, with a total of 1,867,574 tons exported in 2021 [1]. According to McManus et al. [2], Brazilian livestock farming has undergone rapid growth and modernization, mainly in the north (Amazon Biome) and Midwest regions. Freitas Junior and Barros [3] studied the spatial distribution of cattle herds in Brazilian microregions between 1995 and 2016, reporting that herd and cattle densities grew more quickly and significantly in the Amazon regions and that this production growth was particularly due to increased pasture sizes, mainly resulting from deforestation for cattle production. Additionally, animal production intensified, changing production systems due to greater animal density per hectare and consequently changing nutritional, health, reproductive, and facility management [4,5]. An increased number of bovines associated with production intensification directly increases foot diseases that cause considerable economic losses in the beef cattle production chain, such as digital dermatitis, laminitis, heel and sole erosion and ulcers, white line disease, and dewclaw lesions. Foot diseases are estimated to reduce weight gain by up to 25% in sick beef cattle, consequently reducing meat production, productivity, and profit for producers and the agricultural sector [6,7,8,9].

Bovine Dewclaw Dermatitis (BDCD) is characterized by inflammation of the second and fifth accessory digits and the skin in this region. Inflammation is usually associated with necrosis and loss of corneal substance. BDCD onset is associated with a wide variety of risk factors, mainly differences between production systems, geographic location, climate factors, and pastures in different regions and farms, in addition to the physical structures of the pens [5,6,7,8]. BDCD is not a frequently described lesion in the world literature; however, in the Brazilian Amazon Biome region, it has been constantly observed in dairy and beef cattle raised in extensive production systems [10,11,12]. The pathogen has not yet been identified, nor has a pathological diagnosis of BCDC been carried out; therefore, epidemiological studies are extremely necessary. These studies should involve the collection of biological materials so that the epidemiology of BCDC can be elucidated, since the risk factors are generally associated with traumatic dewclaw injuries in cattle raised in recently deforested or cleared pastures with the presence of tree stumps [7].

Almost all studies on hoof injuries are related to dairy herds raised in intensive production systems [3,6,9]. There is a lack of studies on the reality of extensive production systems and beef herds and a lack of data on the incidence and epidemiology of hoof diseases in these animals and production systems, especially regarding dewclaw injuries [10,11,12]. Therefore, the objective of this study was to be a pioneer in describing the clinical diagnosis and the risk factors involved in BDCD onset in beef cattle raised in an extensive production system in the Brazilian Amazon Biome.

## 2. Materials and Methods

The Animal Research Ethics Committee of the Federal University of Pará (CEUA/UFPA) approved this study under protocol n. 080/9838260522. The research was conducted over a year, including the dry and rainy seasons, and the amount of rainfall in the period examined was determined using data from the meteorological station in the immediate area. The study involved approximately ten thousand head of beef cattle from eight farms with extensive breeding systems located in Xinguara, Rondon do Pará, Curionópolis, and Ipixuna do Pará in the south and southeast regions of Pará, Brazilian Amazon Biome. All the farms analyzed had a history of animals with BDCD. A total of 706 Nellore and Nellore crossbred with taurine bovine of both sexes were evaluated, with males aged between 2 and 4 years and a mean weight of 650 kg, and females aged between 2 and 11 years and a mean weight of 400 kg. The distal extremities of bovines with suspected or present accessory digit lesions were inspected during vaccination or reproductive management. These bovines were separated, identified, and restrained in individual containment pens using hoof trimming chutes. Organic matter was removed, and the feet of each bovine sampled were washed with water and neutral soap before undergoing specific clinical examination for lameness, macroscopic lesions and location described according to [13]. The risk factors of BDCD were analyzed according to the following parameters: farm history, pasture characteristics (maintenance, topography, and presence of stones, stumps, and tree trunks), and physical structure of the cattle management facilities [6,7,9,14]. For data analysis, the statistical program IBM SPSS Statistics, version 29.0.2, was used. Initially, the general results of the properties were examined through descriptive statistics and data frequency analysis. The Mann–Whitney U test, also known as the Mann–Whitney–Wilcoxon test, is a non-parametric statistical test used to compare the medians of two independent groups. It is often used when the assumptions of the parametric *t*-test are not met, such as when the data are not normally distributed or when the sample sizes are small.

## 3. Results

Distal extremities of 706 bovines of both sexes were evaluated, with males aged between 2 and 4 years and females aged between 2 and 11 years, with a mean weight of 550 kg. Of these, 506 bovines were of the Nellore breed, and 200 were crossbred (Nellore X taurine breeds), as shown in Table 1. Of the total number of bovines inspected and clinically examined, 49 (6.9%) presented with BDCD (Figure 1 and Figure 2), of which 19 (38.8%) were Nellore, and 30 (61.2%) were crossbred, with 30 (61%) females and 19 (39%) males (Table 2). Also, 15 bovines (30.6%) presented with lesions in more than one dewclaw on different limbs. The pelvic limb dewclaw was affected in 45 bovines (91.8%), with only four bovines (8.2%) affected in the thoracic limbs. We assessed the degree of lameness, and in all assessments, the lameness of the 49 animals with lesions characteristic of BDCD were lameness score 3 (moderately lame) and 4 (lame). From the presumably long duration of existence of these BDCD lesions, it must be deduced that the late presentation of these cases to the veterinarian and/or the neglect of this inflammation in the early stages by the farmers must also be cited as a risk factor that has not yet been mentioned and evaluated in this work but which should be evaluated in future studies.

Of the risk factors identified, five (62.5%) farms (A, E, F, G, and H) had recently cleared or deforested pastures with trunks and stumps, in addition to stones and gravel on the roads used to transport cattle (Figure 3).

A large number of stones was observed in the pastures of six (75%) farms (Table 3), mainly near water troughs and cattle management areas (Figure 4). There was also mud accumulation in the pens, and organic matter adhered to the hooves, especially in the rainy season (Figure 5). The bovines were evaluated in 2022, a year with increased rainfall (September, 57.8 mm; October, 98.6 mm; November, 132.6 mm; and December, 180.79 mm).

The physical facilities used for the bovines were not in adequate maintenance conditions on the farms analyzed, with the presence of broken boards (Figure 6) and exposed nails and screws. There was a high density of bovines in the pens during management, which exceeded the maximum facility capacity, in addition to stressful animal management with the use of sticks and shocks, making the animals visibly agitated.

Farm A had the largest territorial extension, with cattle being transported in vehicles from one area to another due to the long distances between pastures. Trucks with cages were used to transport the bovines. On the floor of these vehicles, we noticed the presence of non-slip iron railings that were not in good condition, not maintained, misaligned, and with loose iron spikes facing upwards (Figure 7).

Foot diseases in animals are not just a health concern; they represent a significant threat to their welfare. These conditions cause immense pain, suffering, and distress, often leading to long-term disability and reduced quality of life. It is crucial to address and prevent these diseases to ensure the well-being and dignity of the animals in our care (see Figure 1, Figure 5 and Figure 6). As in other countries, the animal husbandry conditions currently in place must be recognized not permissible, or even acceptable. At this point, the regions of Brazil, mainly those in the Amazon Biome, immediately need to urgently adapt their practices to improve animal welfare and produce quality, sustainable food that respects the quality of life of animals.

## 4. Discussion

In 1990, the cattle herd in the states that make up the Brazilian Legal Amazon included 26.2 million head. In 2013, this number had increased to 80.7 million head, an increase of 207.38%. The states of Pará and Mato Grosso contributed an increase of 32.3 million head, being responsible for almost 60% of the total increase [15]. This increased agricultural production, especially in cattle farming, is the main cause of deforestation due to the opening of new cattle farming areas, causing serious environmental, social, and cultural damage to the Amazon Biome [16]. Increased cattle farming in recently deforested pastures has also been associated with an increased incidence of hoof injuries in these bovines [7]. The study by Silveira et al. [12] investigated the epidemiological characteristics of foot lesions, reporting that 91.7% (11/12) of the farms had tree-trunk paddocks, 66.7% (8/12) had sloping terrains and stones, and 16.7% (2/12) had flooded areas. These data corroborate the risk factors reported in this study, which include the presence of stones in pastures and roads, mud in management pens, and frequent grazing in recently cleared areas with stumps, stones, and branches. The use of recently cleared areas is a practice followed by most rural farmers in the Amazon Biome, who do not remove fallen vegetation or stumps from these areas before introducing bovines due to the costs, leaving stumps, branches, and roots in the pastures [17]. Therefore, pastures in newly cleared areas increase the onset of traumatic dewclaw lesions caused mostly by stumps, which explains the presence of BDCD in 6.9% of the bovines analyzed in this study. It should be made clear that this figure is likely to be an overestimation when considering all farms in the region as farms with a known history of BDCD were selected for this study.

The Bovines were evaluated in 2022, a year with increased rainfall (September, 57.8 mm; October, 98.6 mm; November, 132.6 mm; and December, 180.79 mm) [18], which is much higher than the expected total for 2023. This explains the accumulation of mud and organic matter in animal management areas on the studied properties, with rainfall and humidity being important risk factors in the Amazon Biome. However, Rodrigues et al. [19] and Klitgaard et al. [20] studied foot disorders in dairy cows, determining spirochetes as the possible agent involved in hoof injury cases, including BDCD, and suggesting that humidity is a relevant factor in BDCD pathogenesis. Humidity was also observed in the present study, especially during the rainy season in which it was conducted.

Another factor evaluated in the present study was the physical structure of bovine management facilities. Costa et al. [21] stated that zootechnical facilities should not cause physical harm to the animals and should ensure the welfare and safety of animals and rural workers. All the farms analyzed in the present study managed animals in pens and containment pens not adapted to best practices aimed at animal welfare, with stressful procedures leading to fear and aggressive behaviors such as restlessness, kicking, agitation, and escape attempts, considerably increasing the risk of digit and dewclaw lesions in the cattle [22,23]. The pens also had containment barriers with broken boards, and access ramps with holes and stones. These features can cause traumatic foot injuries and the consequent onset of BDCD, representing a serious and unacceptable harm to animal welfare. Animal welfare is crucial in beef cattle production for the following reasons: Quality of meat: stress and poor welfare can negatively impact the quality of meat, leading to tougher, less flavorful products, while animals that are well cared for are likely to produce higher-quality meat. Sustainability: Sustainable beef production requires a focus on animal welfare. Providing proper care and management practices can improve efficiency, reduce waste, and minimize environmental impact. Health of animals: Good welfare practices, such as proper nutrition, housing, and disease prevention, contribute to the overall health and well-being of cattle. Healthy animals are more productive and require fewer veterinary interventions. Consumer perception: Consumers are increasingly concerned about the welfare of animals raised for food. Meeting high welfare standards can enhance consumer trust and support for beef products. Ethical considerations: Ensuring good welfare for beef cattle is not just about economic benefits but also about ethical considerations. Cattle have the capacity to experience pain and suffering, so it is our moral responsibility to provide them with a good quality of life. Regulatory compliance: Many countries have regulations and guidelines in place to protect the welfare of farm animals, including beef cattle. Compliance with these standards is essential for legal and ethical reasons. And Brazil, especially for animals raised in the Amazon Biome, needs to comply with these animal welfare standards, so that the country can become not only the largest exporter of meat in the world, but also a quality, sustainable food producer that respects animal welfare. In other words, prioritizing animal welfare in beef cattle production is not only beneficial for the animals themselves but also for the quality, sustainability, and ethical integrity of the beef industry as a whole.

Crossbred bovines were more affected by BDCD (61.2%) than zebu bovines (38.7%). Burrow et al. [24] reported a predominance of zebu breeds and crossings in beef farming, which are considered to be more easily agitated, aggressive, and temperamental than European breeds. Furthermore, human–animal interaction in the extensive farming system is minimal, and with less frequent contact between bovines and rural workers, animals with more agitated temperaments demonstrate more intense and traumatic behavioral responses when exposed to stress situations or human presence [25,26]. Breed, temperament, and extensive breeding systems with reduced management and animal–human contact may have contributed to the agitated behavior of the bovines in this study, which possibly increased BDCD onset.

The physical conditions of the trucks and cages used to transport bovines on farm A may be a predisposing factor associated with dewclaw injuries because the floors of these vehicles had misaligned non-slip iron railings, with loose iron spikes facing upwards, which may increase the risk of injuries during transportation, aspects corroborated by Nielsen et al. [27], who reported that transport-related factors play a major role in the incidence foot injuries in cattle. They reported factors highly relevant to the welfare of cattle during transport based on severity, duration and frequency of occurrence. These included group stress, handling stress, heat stress, injuries, motion stress, prolonged hunger, prolonged thirst, respiratory disorders, restriction of movement, resting problems and sensory overstimulation. The poor maintenance of transport vehicles and the lack of road paving and maintenance cause animal stress during transportation, with less balance in the trucks, which consequently increases the risk of slips and falls, resulting in injuries such as dewclaw lesions.

Rueda et al. [28] also report that frequent and recurrent management in pens can affect regulatory and behavioral responses typical of animals with poor welfare, increasing cattle reactivity and predisposing them to injuries. This factor was observed in the present study, with most animals with BDCD being females (61%) subjected to reproductive management, which requires at least three transportations to the management pen in a short period of time. This greater animal management frequency increases stress levels, making beef cattle used for reproduction more prone and exposed to the risk of accessory digit injuries due to being transported more often to management areas, being subjected to higher stress levels and walking greater distances on soils with stones.

Another factor related to the greater incidence of BDCD in females is age. Molina et al. [29] reported a higher frequency of hoof problems in older bovines which are exposed several times to predisposing factors, reinforcing the results of this present study, in which most bovines with BDCD were older females that stayed for long periods on breeding farms and were more exposed to conditions predisposing them to traumatic dewclaw lesions.

Inappropriate mounting is a behavioral disorder that predisposes digit and accessory digit lesions [30]. Silveira et al. [12] reported corneal tissue erosions in the heels, at the base of the accessory digit, in the pasterns, and in the crown of the pelvic limbs as a result of mounting. In this study, inappropriate mounting was a common practice between females which were intended for breeding on properties 5 and 8 (62.5%) where the fixed-time artificial insemination technique was used. Therefore, on the insemination day, cows in heat constantly mounted the others, predisposing them to foot injuries. In males, mounting was observed during the period they were trapped in the pen before immunization.

Another important factor in digit and dewclaw lesions is parasitism by the fly *Cochliomyia hominivorax*, considered the main species associated with primary myiasis in Brazil [31]. Fly parasitism is facilitated in environments with excess humidity and organic matter and is associated with infrastructure problems and management difficulties which increase the incidence of primary traumatic injuries with hemorrhage that end up attracting flies to deposit eggs in the region [32]. This is a significant factor that cannot be disregarded as a risk factor of accessory digit injuries. Myiasis can be a gateway for foot injury agents, hindering the visualization and diagnosis of primary foot diseases [33]. Diagnosing hoof diseases in beef cattle requires a systematic approach that includes clinical examination, laboratory tests, and sometimes imaging studies. It is also necessary to obtain a thorough history of the animal, including recent changes in diet, environment, or management practices. Any behavioral changes or lameness should be noted. It may be necessary to conduct laboratory analysis of samples, including swabs or scrapings from lesions, to test for bacterial or fungal cultures, for example. Testing for digital dermatitis may involve PCR assays or histopathology. And of course, biopsy of affected tissue may be necessary to confirm the diagnosis, especially if other tests are inconclusive. It is crucial to involve a veterinarian experienced in cattle health for a comprehensive diagnosis. By following these steps, producers can work towards an accurate epidemiological diagnosis of hoof diseases in beef cattle, which is crucial for implementing effective treatment and prevention strategies for BCDC, for example.

Escaping and kicking reactions were observed in the studied bovines, which caused dewclaw injuries, mainly in the pelvic limbs. This kicking behavior was repeatedly observed during management in the pens, resulting in 91.8% of BDCD injuries being observed in the pelvic limbs. This represents another important risk factor for BDCD in beef cattle in the Amazon Biome. The lateral dewclaw (60.60%) was more affected than the medial dewclaw (39.40%), which may be explained by the fact that this structure is more exposed to mechanical trauma [34]. There was greater occurrence of injuries in the right limbs (60.60%). Gargano et al. [35] reported that the right limb is more affected in cattle, possibly due to decreased blood circulation in this limb, since sternal rest commonly exerts greater pressure on this limb.

More studies are needed on this subject to evaluate the actual losses and impacts of BDCD on Brazilian and global agribusiness, as well as studies on the etiology of BCDC in beef cattle farming. It is crucial to address and prevent these diseases to ensure the well-being and dignity of the animals in our care (see Figure 1, Figure 5 and Figure 6). Reducing the occurrence of hoof diseases in beef cattle requires a multifaceted approach that involves both preventive measures and diagnostic strategies. It is essential to implement a schedule for regular inspections of cattle hooves to detect any signs of disease or injury early on. Producers must ensure that the cattle’s living environment is clean and dry, as damp conditions can promote the growth of pathogens that cause hoof diseases. It is necessary to provide a balanced diet rich in essential nutrients to promote overall health and strengthen the hooves. Regular hoof trimming and maintenance can help prevent overgrowth and reduce the risk of injuries and infections. It is important to promptly isolate and treat any cattle showing signs of hoof disease to prevent the spread to other animals. Producers should consult with a veterinarian to develop a comprehensive herd health program that includes vaccination and treatment protocols for common hoof diseases. If an outbreak occurs, tests should be conducted to identify the specific pathogens responsible for the disease. This can help determine the most effective treatment and prevention strategies. Producers should consider breeding programs that focus on selecting cattle with strong, healthy hooves to reduce the likelihood of hoof diseases in future generations. Implementing these strategies can help reduce the occurrence of hoof diseases in beef cattle and improve overall herd health.

## 5. Conclusions

The risk factors analyzed in this study, such as pasture conditions and the physical structure of management facilities, may be responsible for increased BDCD onset in the animals. BDCD affects cattle of both sexes, with varying ages and weights, raised in an extensive system. Stressful and inappropriate management practices are also an important factor associated with BDCD onset since we know the importance of animal welfare in disease prevention. Together with the other risk factors, they should be critical points to be monitored by veterinarians and producers, always focusing on preventive veterinary medicine, to ensure the animals’ welfare, especially concerning hoof infections such as BDCD, in beef cattle in the Amazon Biome.

## Figures and Tables

**Figure 1 animals-14-01329-f001:**
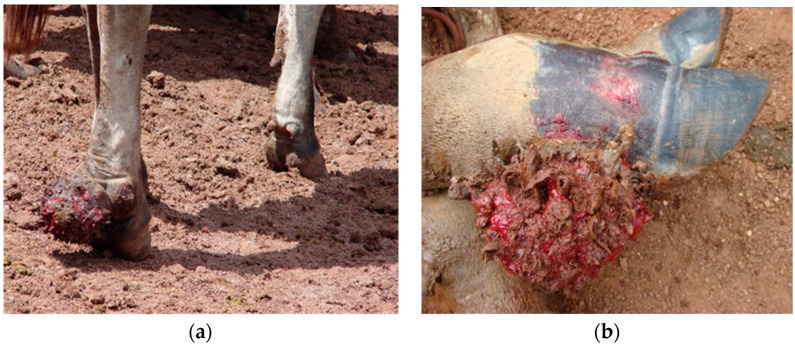
BDCD lesions in beef cattle from farms south and southeast regions of Pará, Amazon Biome. (**a**) BDCD on the left pelvic limb in a pen with stones. (**b**) Chronic BDCD, with corneal tissue loss and increase in volume with proliferation of granulation tissue. Convex surface with filamentous projections interspersed with ulcerated areas and blood clots, on the lateral dewclaw of the right pelvic limb.

**Figure 2 animals-14-01329-f002:**
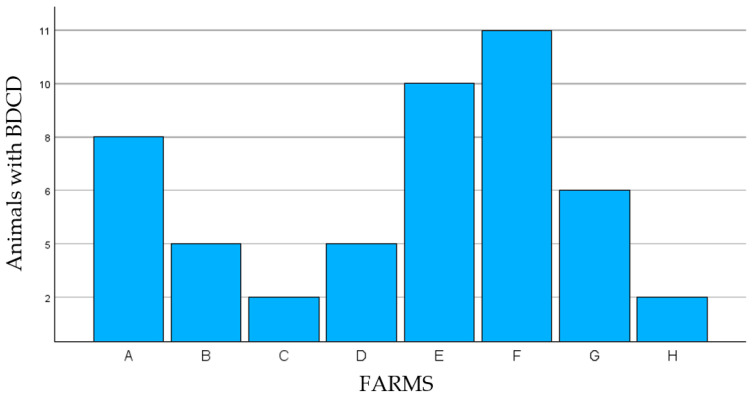
Distribution of BDCD lesions of bovines with BDCD by properties studied.

**Figure 3 animals-14-01329-f003:**
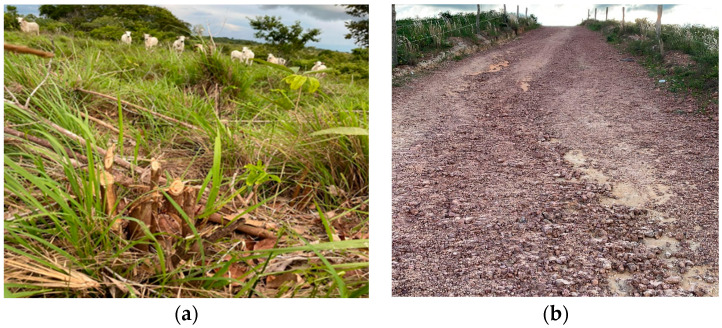
Pasture conditions and roads used for the cattle on farms in south and southeast regions of Pará, Amazon Biome. (**a**) Farm G with cattle in recently cleared pastures with branches and stumps. (**b**) Road with stones and gravel on farm E.

**Figure 4 animals-14-01329-f004:**
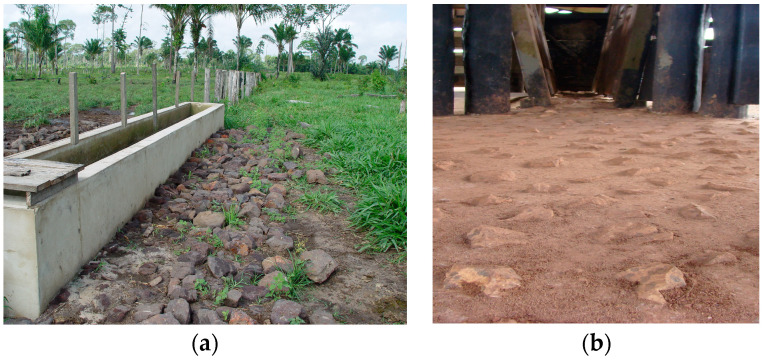
Places where the cattle spent time: (**a**) Farm F with stones around the cattle drinking trough. (**b**) Farm B with stones used in containment pen floor as non-slip objects.

**Figure 5 animals-14-01329-f005:**
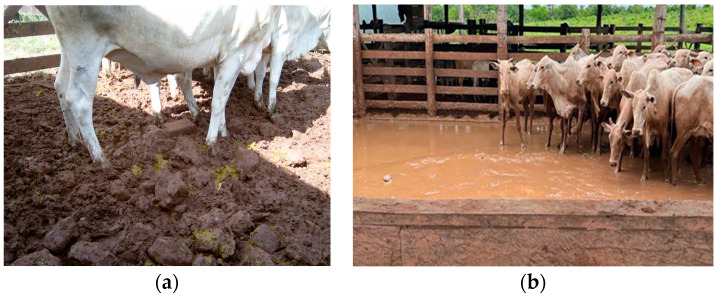
Cattle management pens on the properties studied in the Amazon Biome. (**a**) Management pen with a large quantity of stones on farm D. (**b**) Mud accumulation in a pen during the rainy season on farm C.

**Figure 6 animals-14-01329-f006:**
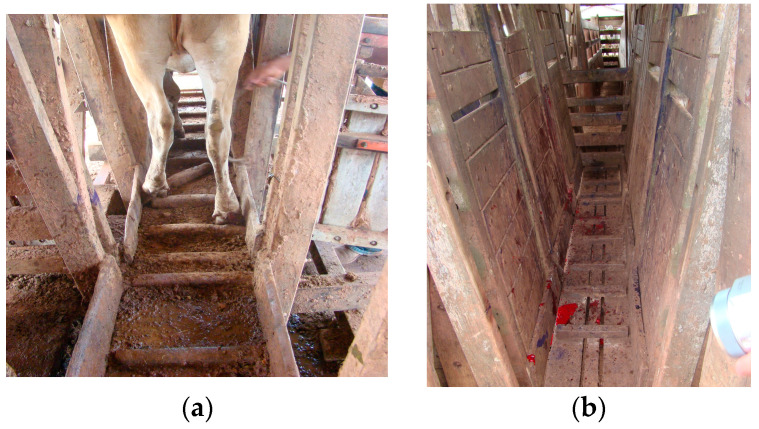
Physical conditions of bovine management facilities on the properties studied in south and southeast regions of Pará, Amazon Biome. (**a**) Individual containment pen on farm B with broken boards. (**b**) Blood on the pen floor, indicating distal extremity injuries in the cattle after management on farm A.

**Figure 7 animals-14-01329-f007:**
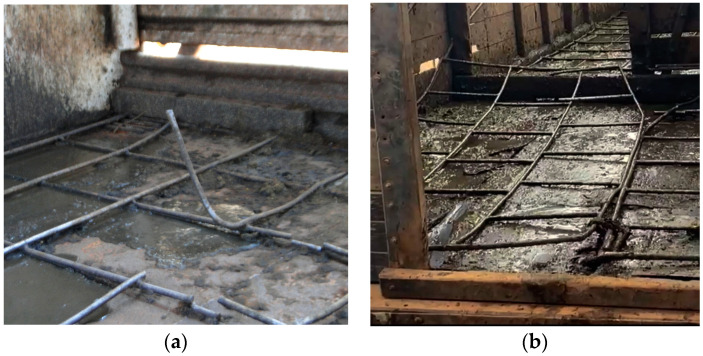
Vehicles used to transport cattle on farm A. (**a**,**b**) Loose transport truck grilles, with misaligned and prominent metal parts with exposed ends.

**Table 1 animals-14-01329-t001:** BDCD lesions in beef cattle raised in extensive systems in the south and southeast regions of Pará, Brazilian Amazon Biome.

Farms	Inspected Bovines	Bovines with BDCD	Limbs		Bovines with BDCD in More than One Limb
			Thoracic Limbs	Pervic Limbs	
A	90	8	1	7	2
B	75	5	1	4	1
C	30	2	0	2	0
D	80	5	1	4	1
E	120	10	1	9	6
F	115	11	0	11	4
G	120	6	0	6	1
H	76	2	0	2	0
Absolute value	706	49	4	45	15
Relative value	100%	6.9%	8.2%	91.8%	30.6%
Minimum	30	2	0	2	0
Maximum	120	11	1	11	6
Average	88.2	6.1	0.5	5.6	1.8
Standard deviation	30.5	3.3	0.5	3.2	2.1

**Table 2 animals-14-01329-t002:** Absolute and relative values of the distribution of dewclaw dermatitis in beef cattle, raised extensively in south and southeast regions of Pará, according to sex, breed and affected limbs.

	Sex		Cattle Breed		Limbs	
	Females	Males	Mestizos	Nellore	Thoracic Limbs	Pelvic Limbs
Absolute value	30	19	30	19	4	45
Relative value	61.2%	38.8%	61.2%	38.8%	8.2%	91.8%

**Table 3 animals-14-01329-t003:** Absolute and relative frequency analysis of the presence of dirty, recently mown pastures and the presence of stones on the studied properties.

		Freshly Mown Pastures		Pastures with Stones		Mann–Whitney Test between Pastures with Stones and Freshly Mown Pastures
		Frequency	Percentage	Frequency	Percentage	
Valid	No	3	37.5	2	25.0	0.2
	Yes	5	62.5	6	75.0	

## Data Availability

Data are contained within the article.

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
