# Peer review of "The Risk Factors of Dewclaw Dermatitis in Beef Cattle in the Amazon Biome"

_animals, 2024, doi:10.3390/ani14091329_

Round 1
Reviewer 1 Report (Previous Reviewer 3)
Comments and Suggestions for Authors
Dear authors,
being involved in the problem of DD in dairy cattle, I have read this paper with more than regular interest. The paper is improved seriously but you should emphasize more that this a serious and unacceptable harm to animal welfare (see Fig 1, 5, 6). another point is that in table 1 and 3 you should maximize your figure behind the comma until 2.
for example, line 299: ... may be an unacceptable predisposing fator ...
Author Response
Dear Reviewer 1,
We appreciate your considerations and the recognition that the article is better than the one presented in 2023. We made the proposed changes, emphasizing the issue of emphasizing more that this a serious and unacceptable harm to animal welfare. We also reduced in all tables and figures, as recommended by you and other reviewers, behind the comma until 1.
Thank you very much!
Sincerely,
Felipe Masiero Salvarani
Reviewer 2 Report (New Reviewer)
Comments and Suggestions for Authors
Overall the paper is simple and clear. The materials and methods needs more detail. My main concern is that there is a heavy focus in this paper about how different factors predispose/ lead to increase in BDCD. However, these links cannot be determined from the statistics carried out in this paper. This would require a detailed risk factor analysis, including random effects, which was not carried out. The proportion of these factors, that you previously deemed important prior to the study, were determined on each farm. This does not show a direct link to BDCD, that that some farms have the factors present.
Abstract
Line 38: There is no evidence from you analysis that these factors are the most important predisposing factors.
Intro
Line 56: Not familiar with the term ‘septic digital dermatitis’, not a common term used in literature on bovine lameness.
M and M
Line 88: How were suspected cases identified? Where there checks every day on each farm to ensure as cattle were identified?
Line 90: Were animals only restrained and inspected at one time point on each farm, or through regular inspections? Where animals treated at this time?
Line 94: Specify in more detail the data collected e.g. what physical structures
Results
Line 133: Graph legend should be below the graph. And this should say ‘Figure’ not ‘Graph’. Provide more detail in the legend.
Line 140: Specific what BPP stands for.
Line 152: Enough space to write: M, F, Mest and Nel in full, rather than using these abbreviations. Will make it easier for the reader.
Line 177: Move text to the left.
Line 183: Include details on statistical test done (Mann-Whitney) in material and methods
Discussion
Line 269: Make it clear that 6.9% if likely over estimated based on all farms in the region, due to farms with known history of BDCD being selected for the study (as stated on line 87).
Line 341: Stick to 1 decimal place for all percentages.
Conclusion
Line 352: There are no statistics done to enable you to say that the epidemiological factor ‘directly increased BDCD’.
Author Response
Dear Reviewer 2,
We appreciate your considerations and regarding your general comment, what we have to justify these risk factors, we understand your concern with our focus on how different factors predispose/lead to increase in BDCD. However, this was the objective of the article, to evaluate whether BDCD actually occurred in the Amazon Biome and what the risk factors for this disease would be, based on data from the literature on hoof diseases in cattle. Although you do not agree with our statistical data, these were inserted, according to requests from former reviewers, who rejected this same article in 2023, so much so that the article now in 2024 was sent with resubmission, that is, everything that led the article to be rejected in 2023 has been corrected and inserted into this current article you reported to. Therefore, I'm sorry, but we do not agree with your statements that "these links cannot be determined from the statistics carried out in this paper" and "this does not show a direct link to BDCD".
Now, point by point, I will answer your questions:
1). Abstract: "Line 38: There is no evidence from you analysis that these factors are the most important predisposing factors". There is indeed evidence, 706 cattle inspected, 49 (6.94%) showed BDCD and this was the first study to determine BDCD's occurrence in extensive farming systems in the Amazon region, also showing that pastures with large amounts of stumps and stones, the physical structure of pens, and trauma and injury incidence during animal management are the most important predisposing factors for the onset of BDCD.
2). Introduction: "Line 56: Not familiar with the term ‘septic digital dermatitis’, not a common term used in literature on bovine lameness." We corrected for digital dermatitis, which is actually the common term used in literature on bovine lameness.
3). Materials and Methods: "Line 88: How were suspected cases identified? Where there checks every day on each farm to ensure as cattle were identified?" The suspected cases were observed on farms had a history of animals with BDCD and the distal extremities of cattle with suspected or present accessory digit lesions were inspected during vaccination or reproductive management. These bovines were separated, identified, and restrained in individual containment pens. Observations on each farm occurred at two times, dry and rainy season, one week on each farm.
"Line 90: Were animals only restrained and inspected at one time point on each farm, or through regular inspections? Where animals treated at this time?" The animals only restrained and inspected at one time point on each farm, at two different times, in the wet and dry season.
"Line 94: Specify in more detail the data collected e.g. what physical structures." A large number of stones were observed in the pastures and the physical facilities used for the cattle were not in adequate maintenance conditions on the farms analyzed, with the presence of broken boards (Figure 5) and ex-posed nails and screws. If necessary, we can insert it into the M&M text, but all this information is found in the results.
4). Results: "Line 133: Graph legend should be below the graph. And this should say ‘Figure’ not ‘Graph’. Provide more detail in the legend." Correction made.
"Line 140: Specific what BPP stands for." The acronym is wrong, it is BDCD and not BPP. Adjusted.
"Line 152: Enough space to write: M, F, Mest and Nel in full, rather than using these abbreviations. Will make it easier for the reader." Modification made.
"Line 177: Move text to the left." Modification made.
"Line 183: Include details on statistical test done (Mann-Whitney) in material and methods." Request included.
5). Discussion: "Line 269: Make it clear that 6.9% if likely over estimated based on all farms in the region, due to farms with known history of BDCD being selected for the study (as stated on line 87)." Inclusion carried out.
"Line 341: Stick to 1 decimal place for all percentages." Modification made.
6). Conclusion: "Line 352: There are no statistics done to enable you to say that the epidemiological factor ‘directly increased BDCD’. " Modification made.
Thank you very much for your suggestions and comments!
Sincerely,
Reviewer 3 Report (New Reviewer)
Comments and Suggestions for Authors
Epidemiology of dewclaw dermatitis in beef cattle in the Amazon Biome
REMARKS AND RECOMMENDATIONS
I have studied as much as possible the literature listed in your bibliography [5, 6, 7, ] on the subject of Bovine Dew claw dermatitis (BDCD). However, I have not found in any of these references a description of the bacteriological examination or a PCR examination for possible etiological infectious bacteria involved, nor have I found any histological examinations which, for example, have proven beyond doubt that these sometimes very large new tissue-formation consists, e.g., mainly by fibrous connective tissue (hypergranulation) and is covered by an ulcerative dermatitis with presence of various bacteria.
In my opinion, no meaningful epidemiological investigations can be started until an etiological cause (histology, bacteriology) of this formation has been clarified and described in detail.
I therefore propose that you carry out exactly these suggested investigations (bacteriological culture tests, PCR for various bacteria and for digital dermatitis specific treponemas, pathohistology, ….) from your tissue samples, which may still be available. Please also check whether BDD-specific treponemes (T. pedis, T. medium, T. phagedenis and others) are involved in these described BDCD-cases. The involvement of these treponemes responsible for bovine digital dermatitis is not unlikely due to the appearance of the superficial ulcerations shown on figures 1a, b.
Without these fundamentally necessary results on the etiology of BCDD, the epidemiological results presented in your manuscript have very little scientific value.
I made several comments and questions in the report below. Please answer these questions and comments and incorporate all my arguments in a completely new and revised manuscript.
INTRODUCTION
Line 60: Here, where you describe this disorder, the reader would expect much more profound information about the type of these large “new” formations originating from the dewclaw (hypergranulated tissue? fibrous connective tissue? Fibroma? tumorous tissue? …) as well as about the type of bacteria that colonize the ulcerated surface of these BDCD-lesions. In the abstract you described that you observed these BDCD-lesions in 49 cattle.
I assume that you took tissue samples from the inner and from the superficial tissue layers of these BDCD-lesions (growths) at the time when you surgically removed these growths. Of these, histological examinations (also with special stains for the identification of BDD treponemes) as well as bacteriological examinations and/or PCR examinations could certainly be carried out in order to check which bacteria play an important role in the pathogenesis of this disease, and which type of tissue makes up the majority of these growths (see e.g.: Pirkkalainen et al. 2024: Local and Systemic Inflammation in Finnish Dairy Cows with Digital Dermatitis. Animals 2024, 14, 461. https://doi.org/10.3390/ani14030461 ).
Line 67: The order of the references in the brackets is not correct: after [5, 6, 7] follows [12, 10]; please check the correct order in the whole manuscript. The reference [8] = Greenough PR. Bovine Liminitis and Lameness. 1. ed. Philladelphia: Saunders Elsevier; 2007. 328p, is missing in the manuscript.
Line 69-71: “BDCD´s epidemiology has not yet been elucidated, being generally associated with traumatic dew-claw injuries in bovine raised in recently deforested or cleared pastures with the presence of tree stumps [7]”. I have studied your reference SILVEIRA et al. 2018 (co-authors of this study), and even in this older study epidemiological aspects of “pododermatitis in the paradigits” were described. Please add here this reference [12] too.
Line 77-78: I strongly recommend that you enlarge the objectives of your study and include the histological and bacteriological (PCR) examination of these BDCD-lesions as the first objective.
MATERIAL & METHODS
You have mentioned more information about M&M in the abstract than here in chapter M&M: there is no information here about the number of cattle examined, how the limbs of these cattle were examined (using hoof trimming chutes?), whether you performed a lameness examination, whether these BDCD-lesions were photographed and whether their size was measured, whether tissue samples were taken and if so, from how many cattle? Please add these requested and necessary information in the chapter of M&M.
In the M&M section you should also describe much in more detail how you investigated these epidemiological factors on the farms; was there a checklist for this? Could this checklist be included in the manuscript or at least linked as additional data sheat that can be queried online?
RESULTS
In the RESULTS chapter I miss the important information about the degree of lameness (locomotion score) in these cattle that suffered from BDCD. You mentioned in the introduction that “foot diseases are estimated to reduce weight gain by up to 25% in sick beef cattle consequently reducing meat production, productivity, and profit for producers and the agricultural sector”. The impact of these foot diseases on economic losses but also on animal welfare is predominantly caused as a result of pain including lameness, that accompanies these (often) painful disorders.
Here in the RESULTS chapter, you should also add the information about the duration of existence of these sometimes very large growths on the dewclaws. Due to the size of this growth (Fig. 1a, b), it must be assumed that several weeks (or even months) have passed since the impacting and triggering trauma occurred until these large growths developed.
From the presumably long duration of existence of these BDCD-lesions, it must be deduced that the late presentation of these cases to the veterinarian and/or the neglect of this inflammation in the early stages by the farmers must also be cited as an epidemiological factor that has not yet been mentioned and evaluated by you.
Is there any information from your rich data set as to whether these BCDD-lesions subsequently recurred after surgical resection? after what time?
Line 127, Fig 1a, b: The ulcerative surface of these BDCD-lesions show a very similar appearance to proliferative BDD-lesions in dairy cattle (see the appendix 1 of the ICAR Claw Health Atlas on page 18, 23, 24, 25: https://www.icar.org/Documents/ICAR-Claw-Health-Atlas-Appendix-1-DD-stages-M-stages.pdf )
DISCUSSION
In the revised discussion, you should first discuss your new histological and bacteriological (PCR) results from samples of these BDCD-lesions, and you should compare these your results, for example, with histological and bacteriological results from other studies on the subject of digital dermatitis (detection of BDD-Treponema spp.) and foot rot (interdigital phlegmon) as well as deep digital sepsis in bovine claws and digits.
You should also discuss the inadequate therapeutic management on the farms by farmers (or veterinarians?), because otherwise these inflammatory BDCD-lesions would not have become so large (as shown in Fig. 1a,b).
Line 270: “The Bovines were evaluated in 2022, a year with increased rainfall (September, 57.8 mm; October, 98.6 mm; November, 132.6 mm; and December, 180.79 mm) [18]”…; You should add this information already in the results section, and furthermore, you should mention in the M&M section, that the mm of rainfall in the period examined, was checked using the meteorological data from the meteo station in the immediate area.
Line 274-277: “However, Rodrigues et al. [19] and Klitgaard et al. [20] studied foot disorders in dairy cows, determining spirochetes the possible agent involved in hoof injury cases, including BDCD, and suggesting that humidity is a relevant factor in BDCD pathogenesis, was observed in the present study, especially during the rainy season in which it was conducted”. Exactly at this point here, your bacteriological / PCR results would be interesting to compare whether you could also detect, e.g., BDD-specific treponemes in these BDCD-lesions.
Line 302-303: “Nielsen et al. [27], who reported that transport-related factors play a major role in the incidence foot injuries in cattle”; You should describe here more in detail which type of foot injuries Nielsen et al. mentioned in their study on transport-related lesions.
Author Response
Dear Reviewer 3,
We would like to thank you for your considerations and in relation to your general comment, what we have to justify is that unfortunately the objective of the work was really to prove whether lesions characteristic of BDCD occurred in the Amazon biome and what the possible risk factors involved in this disease would be. It is not possible for us to do what you advise us to do, "In my opinion, no meaningful epidemiological investigations can be started until an etiological cause (histology, bacteriology) of this formation has been clarified and described in detail. I therefore propose that you carry out exactly these suggested investigations (bacteriological culture tests, PCR for various bacteria and for digital dermatitis specific treponemes, pathohistology, ….) from your tissue samples, which may still be available. Please also check whether BDD-specific treponemes (T. pedis, T. medium, T. phagedenis and others) are involved in these described BCDD-cases. The involvement of these treponemes responsible for bovine digital dermatitis is not unlikely due to the appearance of the superficial ulcerations shown on figures 1a, b." Because the Animal Experimentation Ethics Committee did not provide for the collection of material for etiological research but only for the assessment of the occurrence of lesions characteristic of BCDD and the observation of the risk factors involved. We agree that future work would be important, now with the collection of material for molecular etiological research into the agents involved in BCDD lesions. We know that studies of bovine foot disorders, such as bovine digital dermatitis (BDD), have already demonstrated the involvement of bacteria of the genus Treponema. Given this, we only made the inference that the etiology of BCDD may also be related to bacteria of the genus Treponema, but we do not have data to make such a statement and therefore limit ourselves in discussing this issue of etiology. With the current work, which is the result of a master's degree dissertation, it is intended that the student will now be able to enter the doctorate and then in the phD thesis a new study will be carried out, on more farms, involving metagenomics of the agents and histopathology causing BDCD in cattle and buffaloes in the Amazon biome. I hope you can understand the situation and I propose, if you think it is pertinent, that we change the title to "Risk factors of dewclaw dermatitis (BDCD) in beef cattle in the Amazon Biome".
Now, point by point, I will answer your questions:
1). INTRODUCTION: "Line 60: Here, where you describe this disorder, the reader would expect much more profound information about the type of these large “new” formations originating from the dewclaw (hypergranulated tissue? fibrous connective tissue? Fibroma? tumorous tissue? … ) as well as about the type of bacteria that colonize the ulcerated surface of these BDCD-lesions. In the abstract you described that you observed these BDCD-lesions in 49 cattle. I assume that you took tissue samples from the inner and from the superficial tissue layers of these BDD-lesions (growths) at the time when you surgically remove these growths. Of these, histological examinations (also with special stains for the identification of BDD treponemes) as well as bacteriological examinations and/or PCR examinations could certainly be carried out in order to check which bacteria play an important role in the pathogenesis of this disease, and which type of tissue makes up the majority of these growths (see e.g.: Pirkkalainen et al. 2024: Local and Systemic Inflammation in Finnish Dairy Cows with Digital Dermatitis. Animals 2024, 14, 461. https://doi.org/10.3390/ani14030461 )." As previously mentioned the objective of the work was really to prove whether lesions characteristic of BDCD occurred in the Amazon biome and what the possible risk factors involved in this disease would be.And the Animal Experimentation Ethics Committee did not provide for the collection of material for etiological research but only for the assessment of the occurrence of lesions characteristic of BDCD and the observation of the risk factors involved.
"Line 67: The order of the references in the brackets is not correct: after [5, 6, 7] follows [12, 10]; please check the correct order in the entire manuscript. The reference [8] = Greenough PR. Bovine Liminitis and Lameness. 1st ed. Philladelphia: Saunders Elsevier; 2007. 328p, is missing in the manuscript." Corrections made.
"Line 69-71: “BDCD's epidemiology has not yet been elucidated, being generally associated with traumatic dew-claw injuries in cattle raised in recently deforested or cleared pastures with the presence of tree stumps [7]”. I have studied yours reference SILVEIRA et al. 2018 (co-authors of this study), and even in this older study epidemiological aspects of “pododermatitis in the paradigits” were described. Please add here this reference [12] too." Correction made.
"Line 77-78: I strongly recommend that you enlarge the objectives of your study and include the histological and bacteriological (PCR) examination of these BCDD-lesions as the first objective." As previously mentioned the objective of the work was really to prove whether lesions characteristic of BDCD occurred in the Amazon biome and what the possible risk factors involved in this disease would be. And the Animal Experimentation Ethics Committee did not provide for the collection of material for etiological research but only for the assessment of the occurrence of lesions characteristic of BDCD and the observation of the risk factors involved.
2). MATERIAL & METHODS: "You have mentioned more information about M&M in the abstract than here in chapter M&M: there is no information here about the number of cattle examined, how the limbs of these cattle were examined (using hoof trimming chutes?), whether you performed a lameness examination, whether these BDCD-lesions were photographed and whether their size was measured, whether tissue samples were taken and if so, from how many cattle? Please add these requested and necessary information in the chapter of M& M."Modifications have been made. Some characteristic lesions of BDCD were photographed, but they were not in all animals and no type of measurement of the size of the lesions was carried out. And also, as already explained, there was no tissue sample collection.
"In the M&M section you should also describe much in more detail how you investigated these epidemiological factors on the farms; was there a checklist for this? Could this checklist be included in the manuscript or at least linked as additional data sheat that can be queried online?” The checklist how we investigated these epidemiological factors on the farms, were those already reported in the literature for foot problems in cattle. We wrote down all the factors but there is no way to include this list, as all the factors we investigated are those reported in the work.
3) Results: "In the RESULTS chapter I missed the important information about the degree of lameness (locomotion score) in these cattle that suffered from BCDD. You mentioned in the introduction that “foot diseases are estimated to reduce weight gain by up to 25 % in sick beef cattle consequently reducing meat production, productivity, and profit for producers and the agricultural sector”. The impact of these foot diseases on economic losses but also on animal welfare is predominantly caused as a result of pain including lameness, that accompanies these (often) painful disorders." We assessed the degree of lameness, however the observational study was carried out at two different times, in the rainy season and in the dry season, and in all assessments the lameness of the 49 animals with lesions characteristic of BDCD were lameness Score 3 Moderately lame and 4 Lame . Modification made.
"Here in the RESULTS chapter, you should also add the information about the duration of existence of these sometimes very large growths on the dewclaws. Due to the size of this growth (Fig. 1a, b), it must be assumed that several weeks (or even months) have passed since the impacting and triggering trauma occurred until these large growths developed." We cannot make this statement, since, as mentioned, the observational study took place not continuously, but rather over a week in the rainy season and a week in the dry season.
"From the presumably long duration of existence of these BDCD-lesions, it must be deduced that the late presentation of these cases to the veterinarian and/or the neglect of this inflammation in the early stages by the farmers must also be cited as an epidemiological factor that has not yet been mentioned and evaluated by you." You are indeed right, and in all properties there was negligence on the part of owners, employees and veterinarians regarding the existence of these injuries. I will include this risk factor as something to be observed, but I really cannot say, as it was not evaluated by us.
"Is there any information from your rich data set as to whether these BCDD-lesions subsequently recurred after surgical resection? after what time?" Unfortunately not, because during the study, we indicated to farm owners and veterinarians the treatment to be carried out, but we did not monitor whether the treatment, such as, for example, surgical resection, was actually carried out.
"Line 127, Fig 1a, b: The ulcerative surface of these BDD-lesions show a very similar appearance to proliferative BDD-lesions in dairy cattle (see the appendix 1 of the ICAR Claw Health Atlas on page 18, 23, 24, 25: https://www.icar.org/Documents/ICAR-Claw-Health-Atlas-Appendix-1-DD-stages-M-stages.pdf)." Indeed yes, and that is why some inferences, such as the question of Treponema etiology, were discussed in the work.
4). Discussion: "In the revised discussion, you should first discuss your new histological and bacteriological (PCR) results from samples of these BDCD-lesions, and you should compare these your results, for example, with histological and bacteriological results from other studies on the subject of digital dermatitis (detection of BDD-Treponema spp.) and foot rot (interdigital phlegmon) as well as deep digital sepsis in bovine claws and digits." As previously mentioned the objective of the work was really to prove whether lesions characteristic of BDCD occurred in the Amazon biome and what the possible risk factors involved in this disease would be. And the Animal Experimentation Ethics Committee did not provide for the collection of material for etiological research but only for the assessment of the occurrence of lesions characteristic of BDCD and the observation of the risk factors involved.
"You should also discuss the inadequate therapeutic management on the farms by farmers (or veterinarians?), because otherwise these inflammatory BDCD-lesions would not have become so large (as shown in Fig. 1a,b)." This point was mentioned in the Results, but we cannot make these observations, as we did not carry out this type of analysis, only, as I did in the Results, we can infer that this point of inadequate therapeutic management on the farms by farmers or veterinarians should be evaluated by researchers, so that they have a basis to discuss.
"Line 270: “The Bovines were evaluated in 2022, a year with increased rainfall (September, 57.8 mm; October, 98.6 mm; November, 132.6 mm; and December, 180.79 mm) [18]”…; You should add this information already in the results section, and furthermore, you should mention in the M&M section, that the mm of rainfall in the period examined, was checked using the meteorological data from the meteorological station in the immediate area." Modifications made.
"Line 274-277: “However, Rodrigues et al. [19] and Klitgaard et al. [20] studied foot disorders in dairy cows, determining spirochetes the possible agent involved in hoof injury cases, including BDCD, and suggesting that humidity is a relevant factor in BDCD pathogenesis, was observed in the present study, especially during the rainy season in which it was conducted”. Exactly at this point here, your bacteriological / PCR results would be interesting to compare whether you could also detect, e.g., BDD-specific treponemes in these BDCD-lesions." As previously mentioned the objective of the work was really to prove whether lesions characteristic of BDCD occurred in the Amazon biome and what the possible risk factors involved in this disease would be. And the Animal Experimentation Ethics Committee did not provide for the collection of material for etiological research but only for the assessment of the occurrence of lesions characteristic of BDCD and the observation of the risk factors involved.
"Line 302-303: “Nielsen et al. [27], who reported that transport-related factors play a major role in the incidence of foot injuries in cattle”; You should describe here more in detail which type of foot injuries Nielsen et al. mentioned in their study on transport-related injuries." Modification made.
Thank you very much for all the comments, guidance and questions. I hope that I was able to partially meet your expectations.
Sincerely,
Felipe Masiero Salvarani
Round 2
Reviewer 2 Report (New Reviewer)
Comments and Suggestions for Authors
Authors reviewed comments and made adjustments. I still believe there would be merit in carrying out a sophisticated risk factor analysis, rather than just using descriptive statistics.
Author Response
Dear Reviewer 2,
Thank you for your comments and we believe that your considerations were essential for improving the work, it was just not possible to carry out the sophisticated risk factor analysis.
Sincerely,
Felipe Masiero Salvarani
Reviewer 3 Report (New Reviewer)
Comments and Suggestions for Authors
The authors have taken many of the points from my review into account and incorporated the revised version of the manuscript, which can be seen as positive. However, you have not taken into account the most important point, namely that "...one would expect from a scientific study a profound information about the etiopathogenesis, that means the histologic type of these large “new” formations originating from the dewclaw (hypergranulated tissue? fibrous connective tissue? Fibroma? tumorous tissue? … as well as about the type of bacteria that colonize the ulcerated surface of these BDCD-lesions"; in one word, background information on the etiopathogenesis of this disease.
The authors have justified this by pointing out that "... Because the Animal Experimentation Ethics Committee did not provide for the collection of material for etiological research but only for the assessment of the occurrence of lesions characteristic of BCDD and the observation of the risk factors involved....". I can't really follow this argument because I have already done several clinical studies in which it was necessary to take tissue samples for a histological and bacteriological examination (+ PCR). Given the fact that this sampling was carried out as part of the necessary therapy, in which this entire growth is surgically resected under local anesthesia, the approval of the ethics committee has never been a problem.
Without this etiopathogenetic workup of these conditions in advance, an epidemiological study like this remains of little scientific value.
Author Response
Dear Reviewer 3,
Thank you for your comments and we believe that your considerations were essential for improving the work, the title of the work was changed to just "The risk factors of dewclaw dermatitis in beef cattle in the Amazon Biome" and we make it very clear the need to be new studies were carried out in which epidemiology, with identification of agents and histopathological study of lesions is carried out. Unfortunately, although you state that you do not agree with our justification from the Ethics Committee on Animal Experimentation, I unfortunately need to disagree, as I am part of this committee at my institution and the approved work (CEUA 9838260522 - ID 002023) did not allow the collection of material biological, but just the observation of injuries and risk factors, and I am very ethical in following what I am allowed to carry out in scientific research.
Sincerely,
Felipe Masiero Salvarani
This manuscript is a resubmission of an earlier submission. The following is a list of the peer review reports and author responses from that submission.
Round 1
Reviewer 1 Report
Comments and Suggestions for Authors
The aim of this study was to describe the epidemiological factors involved in the bovine digital dermatitis (BDD), one of the main causes of infectious lameness, in beef cattle raised in extensive production systems in the Brazilian Amazon Biome. The purpose of this work is innovative and important, since the bovine production system in the Amazon Biome has changed with increased cattle density, due to increased pasture sizes, mainly resulting from deforestation, and also because BDD is a debilitating disease not only with economic impact, but also in the animal’s welfare, a major concern nowadays.
General comments
Personally, I don't know if the title of the work “Epidemiology of bovine digital dermatitis (BDD) in beef cattle in the Amazon Biome” reflects its content. What was evaluated - farm history, pasture characteristics (maintenance, topography, and presence of stones, stumps, and tree trunks), and physical structure of the cattle management facilities are farm/herd risk factors associated with the appearance of BDD. I suggest changing the title.
The manuscript is well organized, and easy to read. Materials & Methods are adequate and conclusions are in line with the results obtained. However, Results section could have been presented in more detail, like for instances, showing the number of animals with BDD/farm and the risk factors that were observed.
BDD is a bacterial disease infection that appears to be polymicrobial, with a variety of bacteria, particularly of the genus Treponema, isolated from lesions. I would like to see some discussion relating the bacterial origin of the disease and the risk factors evaluated.
Author Response
Dear Reviewer 1,
Firstly, we would like to thank you for your willingness to review the work and especially for your statement that "work is innovative and important, since the bovine production system in the Amazon Biome has changed with increased cattle density, due to increased pasture sizes, mainly resulting from deforestation, and also because PP is a debilitating disease not only with economic impact, but also in the animal's welfare, a major concern nowadays."
We changed the title as per your request and removed "Epidemiology" and replaced it with Risk factors. Here we would like to make a clarification regarding the translation made about the disease discussed. The work in question studied paradigits pododermatitis (PP) and not bovine digital dermatitis (BDD) as it was in translation. This was an error that has already been corrected in the title and the rest of the article.
Regarding the results, table 1 does present the number of animals with PP/farm, for example, on farm A we selected 90 animals that were suspected of having PP, and of these 90 animals, only 8 were confirmed with PP, one animal with the injury to the thoracic limb and 7 to the pelvic limb. Regarding the risk factors observed, we chose to describe, as shown in lines 113 to 115, "Of the risk factors identified, five (62.5%) farms (A, E, F, G, and H) had recently cleared or deforested pastures with trunks and stumps, in addition to stones and gravel on the roads used to transport cattle (Figure 2)." We believe that the results were very detailed, even including figures to enrich and improve the quality of the Results section.
Regarding the etiology of PP, we did not go into this detailed discussion, as the objective of the article was to first identify whether or not PP occurred in the Amazon Biome. The project forwarded to the Animal Experimentation Ethics Committee did not provide for the collection of material for etiological research but only for the assessment of the occurrence of lesions characteristic of PP and the observation of the risk factors involved. We agree that future work would be important, now with the collection of material for molecular etiological research into the agents involved in PP lesions. We know that studies of bovine foot disorders, such as bovine digital dermatitis (BDD), have already demonstrated the involvement of bacteria of the genus Treponema. Given this, we only made the inference that the etiology of PP may also be related to bacteria of the genus Treponema, but we do not have data to make such a statement and therefore limit ourselves in discussing this issue of etiology. With the current work, which is the result of a master's thesis, it is intended that the student will now be able to enter the doctorate and then in the doctorate a new study will be carried out, on more farms, involving metagenomics of the agents causing paradigits pododermatitis lesions (PP) in cattle and buffaloes in the Amazon biome.
Sincerely,
Felipe Masiero Salvarani

Reviewer 2 Report
Comments and Suggestions for Authors
The authors evaluated the epidemiology of bovine digital dermatitis (BDD) in beef cattle in the Amazon biome, focusing on an understudied topic. They have taken into account how several factors related to environmental conditions directly affect BDD in herds.
Why do you think crossbred (61.22%) have more BDD than Nellore (38.77%)??
Why were the analysis performed with males aged between 2-4 years and female 2-11? Do males not present BDD after the age of 4 years? Could it have been conditioned by age?
Did you survey all the beef cattle present on each farm or did you select a limited number of animals? If you selected the number of beefs cattle for the study, Why did you not select the same number of animals on all farms?
Could the weight loss be associated with a secondary infection caused by these lesions? Are you considering microbiology testing?
Do farms treat injured aniamls with DBB??
Could you add how is the recovery of these sick animals, how long does it take for them to fully recover or if there is no recovery?
Perhaps you can add a paragraph pointing out what would be the best way to treat those animals with DBB and how to solve those pastures characteristics and physical structure to ensure the best confinement of beef cattle.
Limitations: I missed more clinical details about the lesion and, perhaps, some microbiological analysis as the author exposes in the discussion part (Treponema sp and Cochliomyia hominivorax)
Strengths: Innovation in the study of the BDD´s occurrence in farming in the Amazon region. I would like to highlight the length of this study and the work done. In addition, the manuscript is well written and well organized.
Author Response
Dear Reviewer 2,
Firstly, we would like to thank you for your willingness to review the work and especially for your statement that "Innovation in the study of the PP´s occurrence in farming in the Amazon region. I would like to highlight the length of this study and the work done. In addition, the manuscript is well written and well organized."
We changed the title as requested by reviewer 1 and removed "Epidemiology" and replaced it with Risk factors. The title remains: Risk factors of paradigits pododermatitis (PP) in beef cattle in the Amazon Biome. Here we would like to make a clarification regarding the translation made about the disease discussed. The work in question studied paradigits pododermatitis (PP) and not bovine digital dermatitis (BDD) as it was in translation. This was an error that has already been corrected in the title and the rest of the article.
Regarding your questions, we will answer them point by point:
1). Why do you think crossbred (61.22%) have more BDD than Nellore (38.77%)?? Simple, because on the sampled farms the number of crossbred animals was much greater than Nelore PO. 90% of farms in the Amazon Biome raise crossbred animals, but they are animals of zebu breeds and their crosses. In other words, these crossbred animals have Nelore genetics, they are just not pure Nelores (PO).
2). Why were the analysis performed with males aged between 2-4 years and females 2-11? Do evils not present BDD after the age of 4 years? Could it have been conditioned by age? We do not select animals by age. We arrived at the sampled farms and this was the reality of the ages. And these farms only kept male animals up to 4 years of age, as these animals were intended to be raised for slaughter. Unlike females, whose objective was the production of calves and also slaughter. However, as we discussed in the article, age is a risk factor for hoof problems, as according to the literature, a higher frequency of hoof problems in older animals, which are exposed several times to predisposing factors, reinforcing the results of this present study , in which most animals with PP were older females that stayed for long periods on breeding farms, being more exposed to conditions predisposing traumatic accessory digit injuries. And with females in greater numbers and kept longer within the breeding system, they will be more exposed to risk factors for PP, for example.
3). Did you survey all the beef cattle present on each farm or did you select a limited number of animals? If you selected the number of beef cattle for the study, Why did you not select the same number of animals on all farms? We observed all the animals during handling in the corrals, and those that we found to have suspected hoof problems were separated for a specific clinical examination. For example, on farm A we selected 90 animals that were suspected of having PP, and of these 90 animals, only 8 were confirmed with PP, one animal with the injury to the thoracic limb and 7 to the pelvic limb. As there was no selection, it would not be possible to have the same sample number on each of the farms. The sample number varied according to the number of animals with suspected hoof problems.
4). Could the weight loss be associated with a secondary infection caused by these injuries? Maybe so, but that wasn't the aim of the work. Are you considering microbiology testing? The project forwarded to the Animal Experimentation Ethics Committee did not provide for the collection of material for etiological research but only for the assessment of the occurrence of lesions characteristic of PP and the observation of the risk factors involved. We agree that future work would be important, now with the collection of material for molecular etiological research into the agents involved in PP lesions. With the current work, which is the result of a master's thesis, it is intended that the student will now be able to enter the doctorate and then in the doctorate a new study will be carried out, on more farms, involving metagenomics of the agents causing paradigits pododermatitis lesions (PP) in cattle and buffaloes in the Amazon biome.
5). Do farms treat injured animals with DBB? Unfortunately not. Because in the extensive production system, treatment is unfeasible, since the etiology is not known and the predisposing risk factors are often not resolved, or even known by producers and veterinarians, which end up indicating that these animals with hoofs problems are sent for slaughter.
6). Could you add how is the recovery of these sick animals, how long does it take for them to fully recover or if there is no recovery? No. As previously explained, we only spent a few days on each property and did not monitor the progression of the disease, we only recommended, after all, we are researchers and veterinarians, a possible treatment. But the choice of whether or not to treat each of the animals did not depend on our team, but on the rural producer and his private veterinarian. The objective of the work was to identify whether PP occurred on farms and what were the risk factors associated with PP injuries in the Amazon Biome.
7). Perhaps you can add a paragraph pointing out what would be the best way to treat those animals with DBB and how to solve those pastures characteristics and physical structure to ensure the best confinement of beef cattle. The literature demonstrates that the treatment is preventive trim the hoof or surgical treatment. However, this is not the objective of this work. There are already works in the literature that specifically talk about possible treatments for hoof problems. And we believe that by addressing the main risk factors, we are already saying what needs to be done, because by putting photos in the results, illustrating the risk factors, we are already showing the scientific community, rural producers and veterinarians how to carry out prevention. problems, such as animal transport, the quality of pastures and roads, humidity, stones, tree stumps.
8). Limitations: I missed more clinical details about the lesion and, perhaps, some microbiological analysis as the author exposes in the discussion part (Treponema sp and Cochliomyia hominivorax). Regarding the etiology of PP, we did not go into this detailed discussion, as the objective of the article was to first identify whether or not PP occurred in the Amazon Biome. The project forwarded to the Animal Experimentation Ethics Committee did not provide for the collection of material for etiological research but only for the assessment of the occurrence of lesions characteristic of PP and the observation of the risk factors involved. We agree that future work would be important, now with the collection of material for molecular etiological research into the agents involved in PP lesions. We know that studies of bovine foot disorders, such as bovine digital dermatitis (BDD), have already demonstrated the involvement of bacteria of the genus Treponema. Given this, we only made the inference that the etiology of PP may also be related to bacteria of the genus Treponema, but we do not have data to make such a statement and therefore limit ourselves in discussing this issue of etiology. With the current work, which is the result of a master's thesis, it is intended that the student will now be able to enter the doctorate and then in the doctorate a new study will be carried out, on more farms, involving metagenomics of the agents causing paradigits pododermatitis lesions (PP) in cattle and buffaloes in the Amazon biome.
Sincerely,
Felipe Masiero Salvarani

Reviewer 3 Report
Comments and Suggestions for Authors
Dear authors,
Thank you for submitting your manuscript to this journal, however I would qualify this as a case report and not as scientific research. This paper does not include a good statistical estimation of the risk factors you suggest and that is really necessary for acceptation as a research paper.
Some small suggestions compare front and hind feet in any statistical test and season also (e.g. amount of rainfall etc.)
It is interesting that you describe that proliferative inflammation, we hardly do not see that in Western Europe. These cattle is treated topiccaly different as in regular dairy herds is done.
Finally, take more distance of that situations with that broken boards and the transport grills as ethical and from animal welfare not acceptable to work with beef cattle. In our area herds will be closed immediately and cattle must be removed to better housing
In detail: in literature we talking about claw disorder in stead of hoof disease, line 25 and others
Author Response
Reviewer 3:
Firstly, we would like to thank you for your willingness to review the manuscript.
We changed the title as requested by reviewer 1 and removed "Epidemiology" and replaced it with Risk factors. The title remains: Risk factors of paradigits pododermatitis (PP) in beef cattle in the Amazon Biome. Here we would like to make a clarification regarding the translation made about the disease discussed. The work in question studied paradigits pododermatitis (PP) and not bovine digital dermatitis (BDD) as it was in translation. This was an error that has already been corrected in the title and the rest of the article.
Regarding your questions, we will answer them point by point:
1). Thank you for submitting your manuscript to this journal, however I would qualify this as a case report and not as scientific research. This paper does not include a good statistical estimation of the risk factors you suggest and that is really necessary for acceptance as a research paper. No. The article does not fit as a case report, as it is an observational epidemiological study of the main risk factors associated with paradigits pododermatitis (PP) injury in the Amazon Biome. We are not making a statistical comparison of risk factors, but rather demonstrating which ones occur in the Amazon Biome, so statistical analyzes of the frequency of affected limbs (pelvic or thoracic) or even the question of the season are not necessary. The study, I reiterate, was observational, does paradigits pododermatitis (PP) occur or not in the Amazon Biome? What are the risk factors involved? We did not measure which occurred more or less frequently, since each farm sampled had a specific epidemiological situation. And this is the first report in the world of the occurrence of paradigits pododermatitis (PP) in the Amazon Biome, demonstrating the possible risk factors involved.
2). It is interesting that you describe that proliferative inflammation, we hardly do not see that in Western Europe. These cattle are treated topically differently as in regular dairy herds are done. No. The aim of this work is not to describe paradigits pododermatitis (PP), a disease already described in the world literature, but rather to demonstrate that paradigits pododermatitis (PP) occurs in the Amazon Biome and what risk factors are possibly associated. Regarding the etiology or lesions of PP, we did not go into this detailed discussion, as the objective of the article was to first identify whether or not PP occurred in the Amazon Biome. The project forwarded to the Animal Experimentation Ethics Committee did not provide for the collection of material for etiological research but only for the assessment of the occurrence of lesions characteristic of PP and the observation of the risk factors involved. We agree that future work would be important, now with the collection of material for molecular etiological research into the agents involved in PP lesions. We know that studies of bovine foot disorders, such as bovine digital dermatitis (BDD), have already demonstrated the involvement of bacteria of the genus Treponema. Given this, we only made the inference that the etiology of PP may also be related to bacteria of the genus Treponema, but we do not have data to make such a statement and therefore limit ourselves in discussing this issue of etiology. With the current work, which is the result of a master's thesis, it is intended that the student will now be able to enter the doctorate and then in the doctorate a new study will be carried out, on more farms, involving metagenomics of the agents causing paradigits pododermatitis lesions (PP) in cattle and buffaloes in the Amazon biome.
3). Finally, take more distance from those situations with that broken boards and the transport grills as ethical and from animal welfare not acceptable to work with beef cattle. In our area herds will be closed immediately and cattle must be removed to better housing. No. This is the reality in Europe ("In our area herds will be closed immediately and cattle must be removed to better housing), but we need and must demonstrate the reality in Brazil and especially in the Amazon Biome, where today much of the meat is produced exported all over the world. And yes, we need to address hat broken boards and the transport grills as ethical and from animal welfare not acceptable to work with beef cattle, so that changes in Brazilian legislation are made and that important countries question this very important issue which is animal welfare in beef cattle production.
4). In detail: in literature we talk about claw disorder in lieu of hoof disease, line 25 and others. Yes.
As already mentioned here we would like to make a clarification regarding the translation made about the disease discussed. The work in question studied paradigits pododermatitis (PP) and not bovine digital dermatitis (BDD) as it was in translation. This was an error that has already been corrected in the title and the rest of the article.
Sincerely,
Felipe Masiero Salvarani

Round 2
Reviewer 1 Report
Comments and Suggestions for Authors
Dear authors
Thank you for your explanations and for addresing all my comments.
Best regards.
Author Response
Dear Reviewer 1,
Thank you very much for your comments and I am very happy that you understood our explanations and especially that you saw that we accepted your suggestions and modified the work.
Sincerely,
Felipe Masiero Salvarani
Reviewer 3 Report
Comments and Suggestions for Authors
Dear authors,
This is not really an epidemiological study to my opinion, you only present prevelance of foot lesions in your area with a distribution over the breeds. Above that I expected a much more critical attitude of you according the circumstances where these cattle must survive. To my opinion this is not suitable for this Journal others than a critical evaluation.